# Phenylethyl Isothiocyanate: A Bioactive Agent for Gastrointestinal Health

**DOI:** 10.3390/molecules27030794

**Published:** 2022-01-25

**Authors:** Ezequiel R. Coscueta, Ana Sofia Sousa, Celso A. Reis, Maria Manuela Pintado

**Affiliations:** 1CBQF—Centro de Biotecnologia e Química Fina-Laboratório Associado, Escola Superior de Biotecnologia, Universidade Católica Portuguesa, Rua Diogo Botelho 1327, 4169-005 Porto, Portugal; assousa@ucp.pt (A.S.S.); mpintado@ucp.pt (M.M.P.); 2i3S—Instituto de Investigação e Inovação em Saúde, University of Porto, 4169-005 Porto, Portugal; celsor@ipatimup.pt; 3IPATIMUP—Institute of Molecular Pathology and Immunology, University of Porto, 4169-005 Porto, Portugal; 4Medical Faculty, University of Porto, Al. Prof. Hernâni Monteiro, 4169-005 Porto, Portugal

**Keywords:** gastrointestinal health, natural anti-inflammatory, natural antioxidant, natural anti-cancer, nutraceutical, watercress

## Abstract

The incidence of gastrointestinal pathologies (cancer in particular) has increased progressively, with considerable morbidity and mortality, and a high economic impact on the healthcare system. The dietary intake of natural phytochemicals with certain bioactive properties has shown therapeutic and preventive effects on these pathologies. This includes the cruciferous vegetable derivative phenylethyl isothiocyanate (PEITC), a bioactive compound present in some vegetables, such as watercress. Notably, PEITC has antioxidant, anti-inflammatory, bactericidal, and anticarcinogenic properties. This review summarized the current knowledge on the role of PEITC as a potential natural nutraceutical or an adjuvant against oxidative/inflammatory-related disorders in the gastrointestinal tract. We also discussed the safe and recommended dose of PEITC. In addition, we established a framework to guide the research and development of sustainable methodologies for obtaining and stabilizing this natural molecule for industrial use. With PEITC, there is great potential to develop a viable strategy for preventing cancer and other associated diseases of the gastrointestinal tract. However, this topic still needs more scientific studies to help develop new PEITC products for the nutraceutical, pharmaceutical, or food industries.

## 1. Introduction

Over time, gastrointestinal tract diseases, or disorders, have increased in incidence and prevalence, becoming a huge problem for society, with millions of people affected. Gastrointestinal disorders also have a high economic impact and lead to decreased life quality [1,2,3]. Among gastrointestinal tract diseases, the burden of inflammatory bowel disease (IBD) is increasing worldwide. In 2015, in the United States of America, about 3.1 million people were diagnosed with IBD [4]. This disorder is characterized by the non-infectious chronic inflammation of the mucosa, and conditions such as fatigue and weight loss [5]. IBD includes various illnesses, namely, Crohn’s disease and ulcerative colitis, which may later progress to small intestine cancer and colorectal cancer, respectively [6]. In the last few decades, we have witnessed significant progress in cancer research, namely, in its understanding, prevention, and treatment. According to the World Health Organization (WHO), in 2018, cancer was the second principal cause of death globally, with one in six deaths being caused by this pathology [7]. In fact, in a WHO statistical analysis, colorectal cancer was found to be one of the deadliest, corresponding, annually, to 940 thousand deaths, with a mortality rate above 50%.

The WHO estimates that about 30% to 50% of cancers can be prevented by changing or avoiding the main risk factors. Thus, cancer prevention represents an ideal strategy to reduce the burden of this disease, while also offering a more economical treatment strategy in the long term for cancer control [7]. Epidemiological studies established a positive correlation between the increased consumption of vegetables and a reduced risk of chronic degenerative diseases. This was attributed to the wide range of phytochemicals with important physiological properties [8,9]. Among the most promising compounds, phytochemicals from cruciferous vegetables stand out. As a clear example, a compound that is awakening a significant interest is phenylethyl or phenethyl isothiocyanate (PEITC), which, within what has been studied so far, exhibits interesting potential bioactivities (antioxidant, anti-inflammatory, anti-cancer). Indeed, this isothiocyanate is the target of this review article.

Here, we briefly review the current state of knowledge on the implications of PEITC for cancer prevention in the gastrointestinal tract. Simultaneously, we propose a framework to guide the research and development of sustainable nutraceutical solutions for prophylactic action on this global health system’s priority problem. Figure 1 provides a general outline of the ideas put forward for this article.

## 2. What Are Isothiocyanates and Their Natural Sources?

Isothiocyanates are the most abundant products of the natural enzymatic degradation of glucosinolates by the enzyme myrosinase [10,11]. These phytochemical molecules come from cruciferous vegetables (the *Brassicaceae* family), including plants such as watercress, cauliflower, broccoli, and brussels sprouts, among others [11]. Isothiocyanates lead to the characteristic spicy and bitter taste experienced when consuming these plants [12]. To date, a significant number of observational and intervention studies in humans have evaluated the benefits and safety of cruciferous vegetables and isothiocyanate intake [13]. Glucosinolates and their isothiocyanates are compounds proposed to be important contributors to the health benefits of these vegetables, with the anti-cancer effect being one of the main research focuses [13,14]. Phytochemicals from cruciferous plants protect against cancer by modulating the metabolism of carcinogens [13].

The metabolism of glucosinolates can occur with two different types of the stated enzyme: plant myrosinase, which coexists separately with glucosinolates in plants, being activated when the plant is damaged; and bacterial myrosinase, which acts mainly in the colon (comes from gut microbiota) [10]. This metabolism of glucosinolates by gut microbiota can occur when plant myrosinase is denatured. Indeed, the thermal inactivation of plant myrosinase can result in the preservation of some glucosinolates. This is particularly interesting in cooked cruciferous vegetables, as the glucosinolates, when ingested, can be partially absorbed in the stomach, and the remaining intact glucosinolates transit to the colon (due to their hydrophilic nature), where they can be extensively hydrolyzed by the intestinal microbiota and absorbed and/or excreted as isothiocyanates [10,13,14].

Glucosinolates are inert, anionic organic compounds, characterized by β-D-thioglucose, a sulfated oxime group (C=NOSO_3_^−^), and a variable side chain (-R), which will classify the glucosinolate as aliphatic, aromatic, or indole [15]. In response to mechanical or chemical stress, the glycosidic bond of the β-D-thioglucose present in glucosinolates is broken by the enzyme myrosinase, after which this compound converts to isothiocyanates, thiocyanates, and nitriles [16]. The amount of each product formed in this reaction can vary according to the specific proteins, pH, and/or temperature [12,16]. In Figure 2a, the described process is outlined.

Regarding the chemical structure of isothiocyanates, a highly electrophilic carbon establishes two double bonds, one with sulfur and the other with nitrogen, and the radical is linked to the atom of nitrogen. This radical dictates the chemical properties of the active compound [11,17].

The production of isothiocyanates may vary depending on the conditions to which the plant is subjected, namely, the temperature and pH at which the reaction occurs, as well as the availability of ferrous ions and specifying proteins in the medium. Other factors will also determine the reaction’s course and its final metabolites, such as plant species and age, place of cultivation, climatic conditions, storage, and processing [15].

## 3. Phenylethyl Isothiocyanate

Watercress (*Nasturtium officinale*) is a very accessible garden vegetable highly rich in glucosinolates [18]. The most characteristic glucosinolate is gluconasturtiin, an aromatic glucosinolate with an ethyl chain linked to benzene in its radical [19]. PEITC (Figure 2b) results from the hydrolysis of gluconasturtiin by the action of myrosinase. The final product is an isothiocyanate with a phenylethyl radical attached to a nitrogen atom [20].

PEITC is a bioactive compound involved in several biological mechanisms that are naturally related to protecting the plant against external factors. The plant produces PEITC in response to specific stress situations, since it presents biocidal activity against various pathogens, such as bacteria, fungi, insects, and other biotic stressors [11,21]. However, the physiological properties of PEITC are not limited to those exerted at the source of origin. In this way, PEITC can act in humans, combining a series of biological properties with antioxidant, anti-inflammatory, and anti-cancer action. PEITC’s activity on the organism is justified by different bioactive mechanisms, namely, the generation of free radicals, reducing inflammation, and blocking the stages of carcinogenesis [22,23]. PEITC is also known to inhibit cell proliferation, stop the cell cycle, reduce the expression of carcinogenesis, or even tumor suppression via apoptosis and autophagy induction [15,24,25]. Since 2000, PEITC has been one of the main pure glucosinolate derivatives (9.1%) used in clinical trials, particularly to study its anti-cancer effects [10,26].

### 3.1. Antioxidant Action

Oxidative stress is an imbalance between antioxidant and oxidant species, in which the latter prevails over the others, causing an increase in the amount of reactive oxygen species (ROS) in the cell. When ROS concentration is high, damage may occur at the DNA level, increasing the carcinogenesis probability [15]. These ROS come from endogenous sources, such as mitochondrial reactions and cellular inflammatory mechanisms, and exogenous sources, such as exposure to UV radiation and electrophilic molecules [27]. ROS is related to the pathogenesis of diverse gastrointestinal diseases, including gastroesophageal reflux disease, gastritis, enteritis, colitis, associated cancers, pancreatitis, and liver cirrhosis [28].

As a chemo-preventive and antioxidant agent, PEITC may modulate the unregulated ROS concentration in cells, activating antioxidant defense mechanisms through the increased expression of detox enzymes, to lower ROS to basal levels [21,29]. Likewise, PEITC may act as a “selective” antagonist compound for tumor cells, since it acts simultaneously as an oxidizer, inducing ROS production and oxidative damage in tumor cells [17].

### 3.2. Anti-Inflammatory Action

In the gastrointestinal tract, cancer development is closely dependent on energy intake and nutrient availability, but it is also characterized by low-grade inflammation (a slight but chronic increase in the number of various inflammatory markers in the blood and organs) [30]. Inflammation is an immune response caused by several factors, such as infections or tissue damage, which can be subdivided into acute and chronic. Chronic inflammation, which results from an imbalance between pro- and anti-inflammatory cytokines, is closely associated with the pathogenesis of cancer diseases [31].

The excessive expression of a pro-inflammatory factor results in the damage of the epithelial barrier, initiating apoptosis of epithelial cells and the secretion of chemokine. Several researchers have demonstrated the anti-inflammatory properties of PEITC through the reduced expression of this protein by inhibiting NF-κB expression [17,32,33,34]. Pikarsky et al. (2004) and Greten et al. (2004) unequivocally demonstrated that NF-κB plays an essential role in developing liver and intestinal carcinogenesis, respectively [35,36]. In this sense, the prevention of NF-κB activation in hepatocytes was sufficient to inhibit the development of cancers in the livers of mice that were exposed to chronic liver inflammation for seven months [35]. On the other hand, classical colitis-induced carcinogenesis was abolished in mice when targeted to the NF-κB pathway [36]. Therefore, by preventing the activation of NF-κB, PEITC would inhibit cell proliferation and differentiation, and promote apoptosis, leading to cancer prevention.

### 3.3. Anti-Cancer Action

The consumption of the *Brassicaceae* family’s cruciferous vegetables is associated with human health benefits, such as the reduced risk of chronic diseases and several types of cancer, including gastric and colon cancers [11,37]. Based on the literature available so far, PEITC exhibited its anti-cancer effects by inhibiting cell proliferation through cell cycle arrest and tumor cell apoptosis, as well as by resisting metastasis [15,23,24]. Moreover, cancer prevention comes from the effects described above (antioxidant and anti-inflammatory), since its action reduces the risk of developing these pathologies associated with cancer.

Carcinogens are subjected to metabolism and elimination, mainly by phase I and phase II biotransformation enzymes. In general, carcinogenesis occurs due to the bioactivation of carcinogens by phase I enzymes. Hence, phase I metabolism products are highly reactive intermediates that can be harmful by binding to critical macromolecules, such as DNA. In contrast, phase II enzymes play an essential role in the detoxification and excretion of carcinogens from the body. PEITC is involved in the inhibition of phase I enzymes and the induction of phase II enzymes, especially CYP enzymes and transferases, which may explain the chemo-preventive activity [38].

Moreover, it is known that cancer risk can be modified by the dietary intake of bioactive phytochemicals, such as PEITC, which can have epigenetic effects through the modulation of DNA/histone modification. However, currently, studies about the epigenetic mechanisms of these anti-tumor effects in human cells are so far very limited. Indeed, the future development of effective dietary regimes for cancer prevention will require a better understanding of the ingested phytochemicals’ effects on DNA methylation, histone modification, and/or chromatin remodeling in developing tumors [23]. Concerning this topic, Park et al. (2017) reported that PEITC could exhibit chemo-preventive effects and inhibit colorectal cancer progression by inducing pro-apoptotic genes in tumor cells. In particular, PEITC induced stable changes in the tumor cell expression of epigenetic writers/erasers, the chromatin-binding of histone deacetylases, and the hypomethylation of the Polycomb group proteins, as well as other genes that are usually methylated in cancer, which contributed to restricting tumor development [23]. Furthermore, PEITC can be used as an adjunct to increase the potential for other cancer treatments. Giallourou et al. (2019) reported that the compounds extracted from watercress and PEITC improved the therapeutic results of radiotherapy, increasing the DNA damage caused by radiation in cancer cells and protecting non-tumorigenic cells from collateral damage [39].

Despite the encouragement of the foregoing, what is known so far is based on purely empirical studies conducted in controlled laboratory environments. According to the vague epidemiological studies carried out so far, the anticancer properties of PEITC are still under investigation. Up to date, just inverse proportionality correlations have been established between the consumption of PEITC-containing vegetables and the risk of chronic diseases, which makes clear the need to expand the study focus to more in-depth epidemiological studies to have a more complete understanding of the effect of PEITC consumption on cancer relations [15].

### 3.4. Microbial Interaction

Although the antimicrobial capacity of isothiocyanates to control the proliferation of plant and foodborne pathogens has been well documented, currently, there are still few studies that assess the action of these compounds to combat human infections. Furthermore, the information that exists mainly refers to the in vitro antimicrobial activity against bacterial pathogens, and little is known about the in vivo antimicrobial effects of isothiocyanates [14]. In this sense, there is little information about the interaction of PEITC with the microorganisms associated with human gastrointestinal infections and/or commensal flora.

PEITC may act as a potential bactericidal compound against some bacterial pathogens responsible for gastrointestinal infections through the disruption of the plasma membrane, the dysregulation of the enzymatic machinery, and cell death [40]. It is also able to reduce inflammation and inhibit urease activity from *Helicobacter pylori*, blocking its carcinogenic effects in the stomach [11]. These PEITC effects are very important because *H. pylori* infection is difficult to eradicate, and it is a major cause of gastritis and peptic ulcers, a condition that may be associated with the development of gastric cancer [14]. Concerning the antibacterial activities of PEITC against harmful intestinal bacteria, this phytochemical can strongly inhibit the growth of *Clostridium difficile* and *Clostridium perfringens*, two pathogenic agents of the genus *Clostridium* that can threaten human health [14,41]. Thus, it would be interesting to develop further studies to describe the potential clinical efficacy of PEITC as a therapeutic or preventive agent for the treatment of diseases caused by harmful gastrointestinal bacteria.

Furthermore, it is of particular interest to understand its interaction with the intestinal microbiota. Isothiocyanates, as products of myrosinase−based glucosinolate hydrolysis in the human gut, are important to health, particularly their anti-cancer properties and other beneficial roles in human health mentioned above [42]. Even though this is an emerging topic, the research studies on the impact of glucosinolates and their isothiocyanates on gut microbiota are still very scarce. However, a close relationship between the consumption of glucosinolates, their metabolism, and the intestinal microbiota composition has been suggested [10]. Indeed, in vitro studies have demonstrated the potential of *Bifidobacterium* sp., one of the common bacteria belonging to the human intestinal microflora, in the hydrolysis of glucosinolates [10,43]. Kellingray et al. (2017) examined whether a *Brassica*-rich diet in healthy adults was associated with changes in the gut microbiota composition. The study concluded that a diet rich in *Brassica* did not significantly alter the relative proportions of intestinal lactobacilli, but was associated with a reduction in the relative abundance of SRB [44]. Further studies about bacterial strains involved in the degradation of glucosinolates in the colon, the characterization of degradation products (particularly isothiocyanates), and their physiological effects on the intestinal microbiota are needed to understand the modulation of the gut microbiota by the metabolites of the cruciferous vegetables [43,44].

According to Kim and Lee (2009), PEITC showed not to have an antimicrobial action against commensal bacteria (*Bifidobacterium bifidum*, *Bifidobacterium breve*, *Bifidobacterium longum*, *Lactobacillus acidophilus,* and *Lactobacillus casei*), and thus did not negatively affect the intestinal microbiota [41]. This opens potential opportunities to explore whether these compounds can also contribute to health by interacting with the gut microbiota through several pathways, such as prebiotics.

In addition, PEITC exhibits strong antimicrobial potential against pathogens that compromise food safety, an essential public health issue that continues to be a significant concern for consumers, regulators, and food industries worldwide [45]. PEITC can interact with cell surface constitutes and consequently compromise the integrity of the cytoplasmatic membrane of bacteria, causing foodborne diseases such as *Escherichia coli*, *Listeria monocytogenes*, *Pseudomonas aeruginosa*, *Staphylococcus aureus*, and *Vibrio parahaemolyticus* [45,46,47].

## 4. Biocompatibility of PEITC

Currently, clinical trials have been carried out to prove the chemo-preventive properties of PEITC, for which it was necessary to analyze the toxicity in humans in order to establish limit values. In a study conducted in 2018, a dose of 40 to 80 mg of PEITC per day was administered to humans orally for 30 days, with no adverse effects [29]. However, when the dose consumed was increased to values between 120 and 160 mg of PEITC per day, during the same experimental period, some toxicity was observed, although it was not lethal [29]. Other authors concluded that the acceptable daily dose is only 40 mg in humans, which is consistent with the study previously presented [48]. Thus, there is still no definitive range of concentrations for levels of toxicity in humans. However, it is known that after specific doses, PEITC can be considered to be toxic and interact with other drugs that are being taken simultaneously. According to Abbaoui et al. (2018), for therapeutic effects, non-toxic doses of PEITC are sufficient, being safe for human consumption [16].

## 5. Dosages of PEITC

Knowing the dosages of a product is essential for its formulation, since it is essential to know which quantities should be ingested for a given purpose. In this article, we analyzed two aspects of the application in which the dose–effect is different, leading, on the one hand, to the prevention of pathologies that culminate in more severe problems, such as cancer, and, on the other hand, to the therapeutic effect of PEITC. To assess the best way forward, PEITC dosages for different cancer types were tested out on cell lines. Thus, the effects caused by tumor cells when exposed to different concentrations of PEITC were studied. A range of PEITC performance values was also established, from the more preventive to the most curative phase, ranging from 5 to 30 μM, corresponding to 0.82 to 4.90 mg L^−1^. This information is gathered in Table 1.

Thus, we can establish that an effective dose for prevention corresponds to a concentration of PEITC between 5 and 10 μM, equivalent to between 0.82 and 1.63 mg L^−1^, since these dosages cause an anti-inflammatory action and the inhibition of cell proliferation. For PEITC to have a more therapeutic effect, the dosage must be higher. A concentration between 10 and 30 μM, which corresponds to between 1.63 and 4.90 mg L^−1^, would be the most suitable, since there is an induction of apoptosis and a loss of viability for tumor cells.

As we have already established, in this article, the analysis of the state of the art on prophylactic and therapeutic properties of PEITC is focused on the gastrointestinal tract. In this sense, the most information on the chemo-active properties of PEITC comes from the study of colon cancer, being almost nil for gastric cancer. However, most of the studies are only at exploratory stages in vitro, which clarifies the need for progress to other phases to validate the findings.

## 6. PEITC Extraction

We have already discussed what is known so far about the potential properties of PEITC to prevent gastrointestinal disorders. PEITC is also a natural and sustainable compound, since it can be extracted from the by-products of watercress, a raw material that is not crucial to human needs [53]. This reasoning guarantees the preservation of ecosystems, allowing the label of “environmentally friendly” to be affixed. Additionally, it has a very high dose–effect relationship, since for the PEITC to have a preventive action, the necessary dose can be obtained through a tiny amount of watercress, which can be considered to be an economically viable process.

So far, the reported work on obtaining PEITC is scarce. Furthermore, few studies apply a sustainable approach [53,54,55,56]. The applied methodologies, in general, are analytical, and use polluting organic solvents that are not feasible at an industrial level, even more so when considering their use in the gastrointestinal tract. The few works with sustainable methodologies apply more complex and expensive techniques and are not easily scalable, e.g., the use of microwave-assisted ethanol extraction or supercritical fluids [37,54,57]. In Table 2, works with different methodologies and their advantages/disadvantages are reported. Through the analysis of Table 2, it appears that the extraction with aqueous micellar systems with non-ionic and biodegradable surfactants is the most advantageous. With this extraction method, the final product has no additives, does not contain toxic products, and is extracted sustainably. Likewise, it is a technique that needs to be further explored and optimized to increase its yield, with, for example, its incorporation in the process of external myrosinase to increase the conversion of glucosinolates, as was performed with the other methodologies. Simultaneously, by observing Table 2, it is possible to establish a range of values for PEITC in mg per 100 g of fresh watercress, between 10.5 and 68.8 mg.

## 7. Conclusions and Future Perspectives

In summary, PEITC, a product of glucosinolate hydrolysis found in cruciferous vegetables, has been extensively studied for its preventive and therapeutic effects in chronic diseases due to its antioxidant, anti-inflammatory, and anti-cancer properties. Moreover, PEITC can exhibit antibacterial activity against harmful bacteria in the gastrointestinal tract. Therefore, studying this compound as a potential natural antimicrobial agent against human infections might be interesting. Moreover, this topic is a promising area of study, especially considering the need to develop new antibacterial products, since drug-resistant infections are a significant threat to people’s health [14]. Another emerging topic also includes the effects of PEITC on gut microbiota interactions, in line with the growing study of the gut–brain axis, which different bioactive compounds can modulate in different ways. This opens an opportunity for further investigation, as there is still a lot of lack of information on the subject.

The bioactive properties of PEITC are still under investigation, which creates a need for clinical studies to prove the safety and effectiveness of PEITC in humans. After validating its biological properties in vivo, PEITC could achieve promising integration in the pharmaceutical industry. Thus, the use of PEITC as an adjunct to existing medication for cancer treatment would reduce the amounts of drug administered, thus reducing its side effects [60]. Even so, for gastrointestinal disorders PEITC performance is underexplored. Its in-depth exploration would help the development of new products for the pharmaceutical and food industries with their nutraceutical lines.

Even though PEITC has great potential as a health-promoting compound, its industrial use has been limited because of its relative instability [56]. PEITC is a highly reactive electrophile, susceptible to attack by nucleophilic molecules [61]. Furthermore, PEITC is a compound with low molecular weight (MW = 163.2 g mol^−1^) and considerable hydrophobicity (logP = 3.47). Its pharmacokinetic features include first-order linear absorption with a high protein binding nature [62]. Therefore, its stabilization becomes a technological challenge. An option to stabilize PEITC, and even increase its bioavailability in a food matrix, is micro/nanoencapsulation. However, PEITC and ITCs micro/nanoencapsulation has been poorly studied [63]. Till now, cyclodextrin and chitosan microparticles were reported to be plausible carriers for isothiocyanates [64,65]. Besides, PEITC was already stabilized with vegetable oils that protect non-polar isothiocyanates from decomposition or volatilization [56]. This opens a relevant research line aiming to identify the process conditions that could be used at an industrial level and explore or design different food and pharmaceutical matrices in which it can be incorporated. That is why the greatest opportunities for progress in the field are found in the search for alternatives, and the optimal stability of PEITC through different strategies for its use in the most diverse products remains possible. It is important to note that, in the pharmaceutical market, PEITC-based products are non-existent, while in the nutraceutical market, we hardly find low-purity watercress extracts. Therefore, after filling these gaps that still exist, an innovative strategy to respond to market needs would be the development of PEITC-based products associated with its biological properties as a preventive agent or as an adjuvant to existing treatments for cancer.

## Figures and Tables

**Figure 1 molecules-27-00794-f001:**
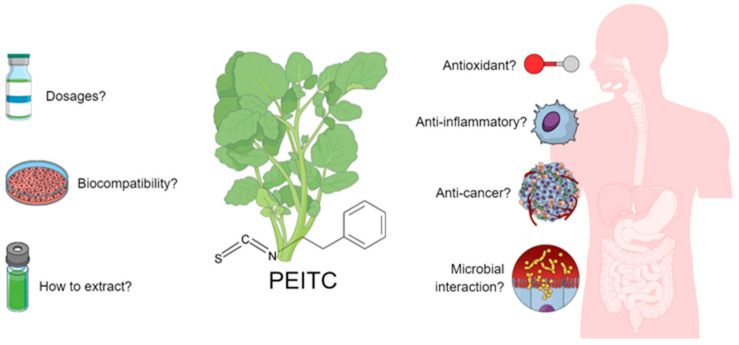
Outline of the general ideas for the literature review.

**Figure 2 molecules-27-00794-f002:**
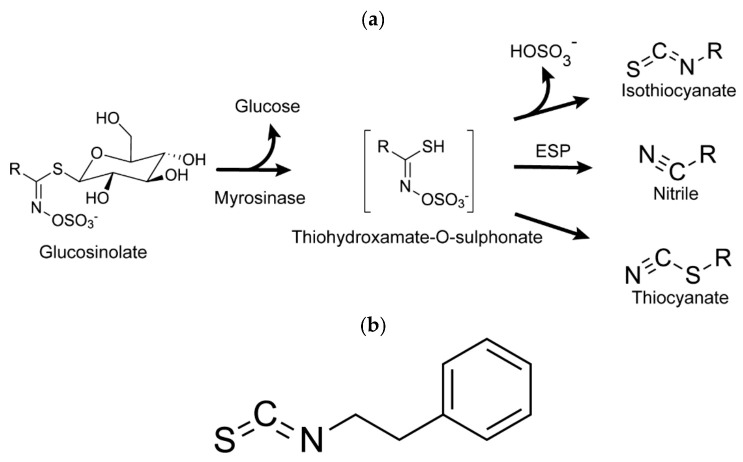
Glucosinolate hydrolysis and PEITC. Glucosinolate reaction catalyzed by the enzyme myrosinase (**a**). Phenylethyl isothiocyanate (**b**).

**Table 1 molecules-27-00794-t001:** PEITC doses for different types of cancer and their mechanisms of action.

	PEITC ^1^	PEITC ^2^	Effect	References
Colon cancer	10.0	1.63	Attenuation of inflammation and cell proliferation	[17]
	10.0–40.0	1.63–6.53	Suppression of cell proliferation and loss of viability of tumor cells	[49]
			Apoptosis and anti-inflammatory action	[50]
	10.0	1.63	Tumor regression	[38]
	2.5–15.0	0.4–2.45	Inhibition of proliferation	
	1.0–5.0	0.16–0.82	Apoptosis	
	10.0	1.63	Anti-inflammatory action	
Gastric cancer	1.5	0.24	Apoptosis	
Cervical cancer	5.0–10.0	0.82–1.63	Cell proliferation inhibition and apoptosis induction	[24]
	15.0	2.45	Apoptosis	[51]
Breast cancer	20.0–30.0	3.26–4.90	Inhibition of cell proliferation and cell cycle arrest	[39]
Prostate cancer	5.0–7.5	0.82–1.22	Decreased expression of the NF-kB factor (anti-inflammatory action)	[50]
Lung cancer	12.5–20.0	2.04–3.26	Cell cycle arrest and apoptosis	
Laryngeal carcinoma	0.0–10.0	0.00–1.63	Inhibition of cell growth, cell cycle arrest, and apoptosis	
Leukemia	4.0	0.65	Beginning of apoptosis	[52]
	6.0–8.0	0.98–1.31	Significant increase in apoptosis	

^1^ Values in μM. ^2^ Values in mg L^−1^.

**Table 2 molecules-27-00794-t002:** Advantages and disadvantages of the PEITC extraction methods.

Extraction Method.	Amount of Extracted PEITC ^1^	Advantages	Disadvantages	References
Aqueous micellar systems with autolysis	10.5–14.0	It does not involve toxic solvents; reduced cost; sustainable; stabilized PEITC; “clean label” product	Depends on the amount of endogenous myrosinase present in the watercress	[53]
Organic solvent	23.3–68.8	Direct and ready-to-use technique; reduced costs	Toxic organic solvents; addition of external myrosinase; loss of active compound through filtration and evaporation	[57,58,59]
Pressurized fluid	33.5	A higher amount of extracted PEITC; does not involve toxic solvents; preserves the bioactivity of the compound	The use of high pressures; requires more sophisticated equipment	[37,57]

^1^ Values in mg PEITC 100 g^−1^ fresh watercress.

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
