# Peer review of "Phenylethyl Isothiocyanate: A Bioactive Agent for Gastrointestinal Health"

_molecules, 2022, doi:10.3390/molecules27030794_

Round 1

Reviewer 1 Report

  1. In Abstract "PEITC antioxidant, anti-inflammatory, bactericidal, anti-cancer properties are of particular importance"  Re-wright this sentence.
  2. Line no 64 "I" should be in capital letters.
  3. Line no 66 Please explain in short how glucosinolates and their isothiocyanates are compounds contributors to the health benefits of these vegetables. 
  4. If possible then add a figure that describing the whole review. (Just like a graphical abstract).

Author Response

We thank the reviewer for the positive feedback, as well as for the comments that helped to improve the article. Each of the suggestions was considered and can be found in the new version of the manuscript, marked by the Track Changes in Microsoft Word.

Author Response

(The authors gave the same response as above.)

Reviewer 3 Report

see attachment

Author Response

(The authors gave the same response as above.)
